# Hidden pressurized fluids prior to the 2014 phreatic eruption at Mt Ontake

Corentin Caudron [1,2] ✉, Yosuke Aoki [3], Thomas Lecocq [4], Raphael De Plaen[4], Jean Soubestre [5], Aurelien Mordret [2], Leonard Seydoux [2] & Toshiko Terakawa [6]

A large fraction of volcanic eruptions does not expel magma at the surface. Such an eruption occurred at Mt Ontake in 2014, claiming the life of at least 58 hikers in what became the worst volcanic disaster in Japan in almost a century. Tens of scientific studies attempted to identify a precursor and to unravel the processes at work but overall remain inconclusive. By taking advantage of continuous seismic recordings, we uncover an intriguing sequence of correlated seismic velocity and volumetric strain changes starting 5 months before the eruption; a period previously considered as completely quiescent. We use various novel approaches such as covariance matrix eigenvalues distribution, cutting-edge deep-learning models, and ascribe such velocity pattern as reflecting critically stressed conditions in the upper portions of the volcano. These, in turn, later triggered detectable deformation and earthquakes. Our results shed light onto previously undetected pressurized fluids using stations located above the volcano-hydrothermal system and hold great potential for monitoring.

The 2014 eruption at Mt Ontake in Japan (Fig. 1) highlighted our limited ability to anticipate small non-magmatic eruptions. In a country optimally equipped to face natural disasters, the eruption claimed the life of more than 50 hikers, becoming the worst volcanic disaster in Japan in almost a century. These non-magmatic eruptions, so-called phreatic, hydrothermal[1], hydro-volcanic[2] or steam-driven eruptions[3], do not release juvenile material but remain among the most difficult to forecast, therefore representing a potential risk. The absence of forecasting signals[4] challenges volcanologists' knowledge. Despite their frequent occurrence, they remain poorly understood and have recently caused human fatalities; the 2019 White Island eruption in New Zealand also killed more than 20 people[5]. It is therefore paramount to better investigate such eruptions by integrating new methodologies to fully understand the preparatory processes at play and improve our ability to forecast these eruptions.

The Mt Ontake eruption was initiated without any clear surface precursor[6]. It consisted of three phases starting with pyroclastic density currents flowing 2.5 km away from the craters, followed by increased ash injection into the atmosphere and ejection of ballistics, and finally, muddy hot water flowing from the craters[7]. Following the eruption, numerous studies focused on detecting potential precursors. In retrospect, useful precursors were identified seconds to minutes before the eruption (e.g., Kato et al.[8]), while long-term changes in spring water compositions and ground level were also documented (e.g., Sano et al.[2]). The former would be identified too late in issuing a successful warning while the long-term precursors are based on data too sparse to be useful for near-real-time forecasting[6]. Some mid-term precursors (days to 1 month) were highlighted by re-examining geodetic[9] and seismic data[10]. The ground uplift from geodetic data remained ambiguous, whereas the deviation of the local

[1]Université Libre de Bruxelles, Brussels, Belgium. [2]Univ. Grenoble Alpes, Univ. Savoie Mont Blanc, CNRS, IRD, Univ. Gustave Eiffel, ISTerre, Grenoble, France. [3]Earthquake Research Institute, The University of Tokyo, Tokyo, Japan. [4]Seismology- Gravimetry Department, Royal Observatory of Belgium, Uccle, Belgium. [5]Instituto Volcanológico de Canarias (Involcan), Granadilla de Abona, Tenerife, Spain. [6]Nagoya University, Nagoya, Japan. ✉e-mail: corentin.caudron@ulb.be

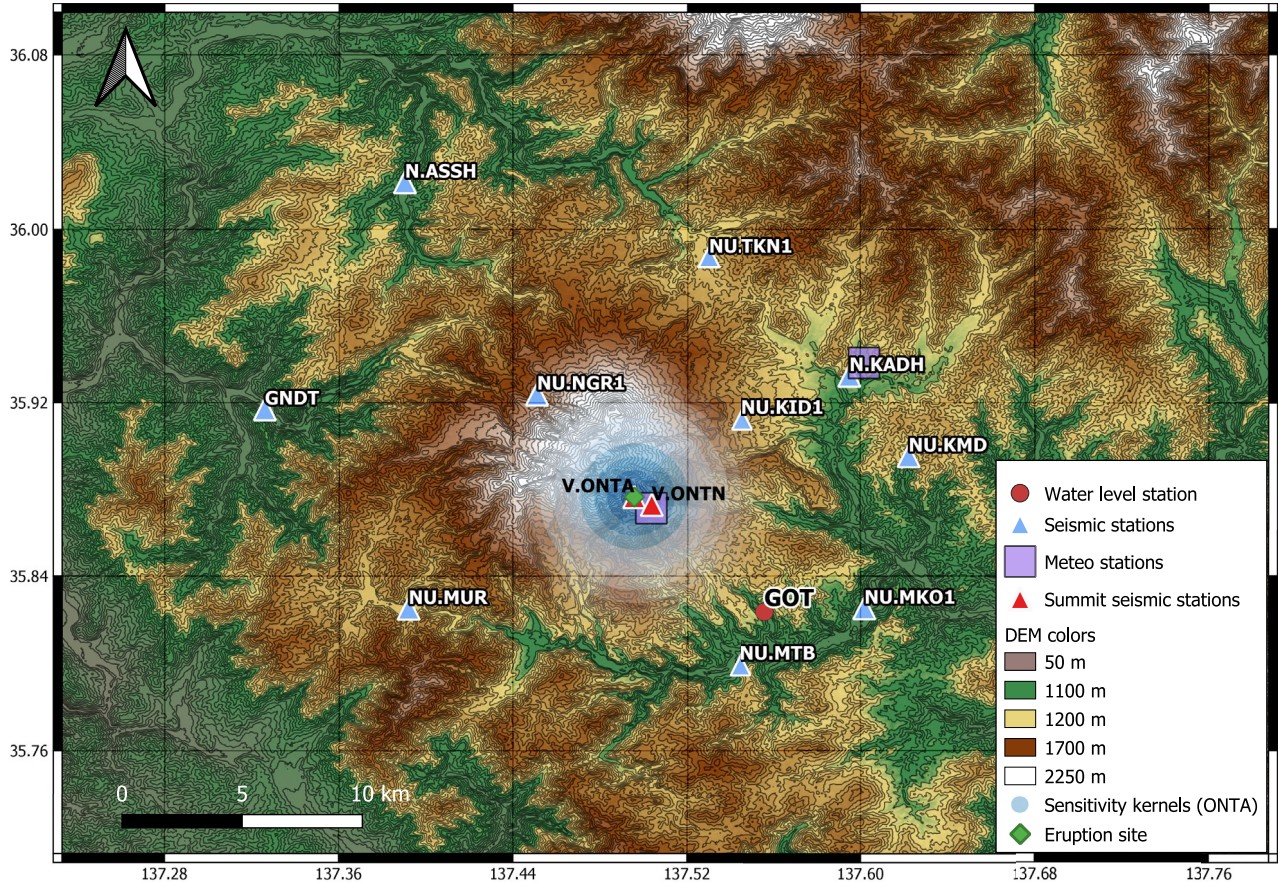

**Fig. 1 | Map of Mt Ontake network.** Seismic station locations (triangles). MKO and TKN are broadband seismometers (natural period of 120 s), whereas the others are short-period sensors (natural period of 1 s). The locations of the meteorological stations (purple transparent squares co-located with seismic stations N.KADH and V.ONTN) and the groundwater station (red circle; the reference water level is 645 meters below the surface) are also indicated. The eruption site is shown as a green diamond. The contour lines and shadings are created from an Aster Digital Elevation Model. The sensitivity kernels for station ONTA are shown as transparent white and blue circles.

stress field from seismic data[10] would practically be difficult to implement in most volcano observatories. It would also require sustained earthquake activity. The forecasting and precursory processes of this eruption, therefore, remain difficult to untangle[6].

Recent advances in volcano monitoring took advantage of seismic interferometry to better monitor volcanoes (e.g., Sens-Schönfelder and Wegler[11], Brenguier et al.[12], Donaldson et al.[13,14], Yates et al.[15]). This technique has allowed detecting tiny changes in seismic wave propagation prior to (Brenguier et al.[12]) and during magmatic eruptions (Donaldson et al.[13]). The analysis of continuous seismic time series also has recently seen a breakthrough due to the development of machine-learning-based strategies. While the classification of seismo-volcanic signals is performed routinely by analysts, recent studies have trained models to automatically classify waveforms (e.g., for volcanic tremors, volcano-tectonic earthquakes or long-period events[16,17]). In such approaches, the models are supervised, i.e. the models are trained to recognize events from a set of already labeled waveforms. Several other studies have also developed unsupervised strategies such as waveform clustering[18,19]. These other approaches allow to define classes with no a priori and are generally preferred when the dataset is largely unlabeled or when the number of classes cannot be easily inferred. Among the different deep-learning earthquake detectors, several algorithms allow to pick the phases of earthquake-generated signals[20,21] and hence to invert the position of the hypocenters in space.

In the present study, we first apply the most recent developments in seismic interferometry[22–24] to check for seismic velocity variations before the 2014 eruption. In addition, we perform a systematic detection of the sources in action by quantifying the degree of spatial coherence from the continuously calculated cross-correlations (as in Seydoux et al.[25] and Soubestre et al.[26]). Finally, we use the most recent deep-learning earthquake detector *EarthquakeTransformer* in a predictive mode directly, by considering the model trained on other datasets[27].

## Results and discussion

### Seismic velocity variations

In this study, we computed daily single-station inter-component auto-correlations (ACF) and inter-station single-component cross-correlations (CCF) at 11 seismic sensors of the Ontake network (Fig. 1) to estimate seismic velocity changes in the shallow crust, following standard processing similar to Lecocq et al.[28] and De Plaen et al.[29]. The velocity variations, dv/v, are calculated from 5-day stacks of ACF and compared with the full period stack as reference.

At sensors located on the summit (red triangles in Fig. 1), we observe a velocity drop from June to October 2014 in the 1–2 Hz frequency band (Fig. 2 for station ONTA). After the eruption, seismic velocity kept decreasing for a few days. Such drop was only recovered at seismic stations located on the summit of the volcano (Fig. 1) and in the 1–2 Hz frequency band (Fig. 2). Various processing parameters and data processing approaches have been tested (see methods and supplementary materials for details) and all gave similar results.

We then attempted to better constrain the nature of this observation through post-processing of dv/v observations. Non-volcanic perturbations can decrease crustal seismic velocities, as recently

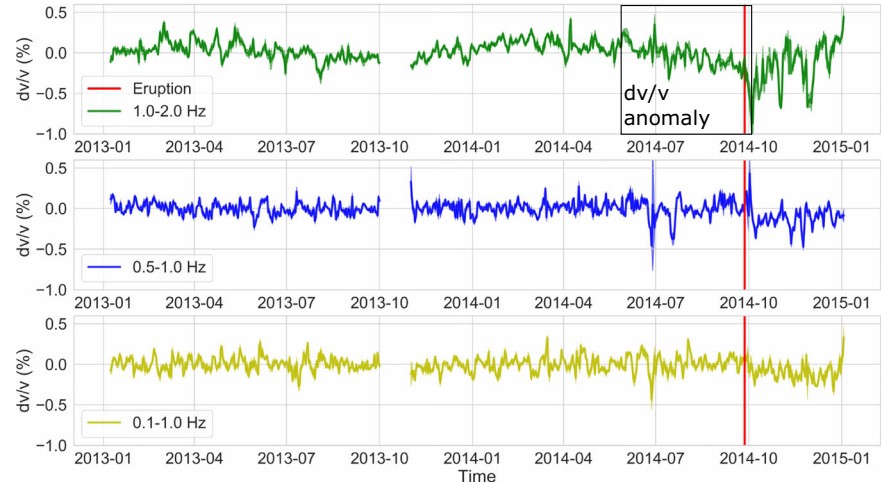

**Fig. 2 | Relative velocity variations from AC.** Temporal evolution of relative seismic velocity at station ONTA using different frequency bands (1.0–2.0, 0.5–1.0, and 0.1–1.0 Hz). The vertical red line indicates the 27 September 2014 eruption.

Errors are shown as shaded values around each curve and are estimated following Lecocq et al.[28]. A velocity anomaly is observed from June to October 2014 in the 1–2 Hz frequency band.

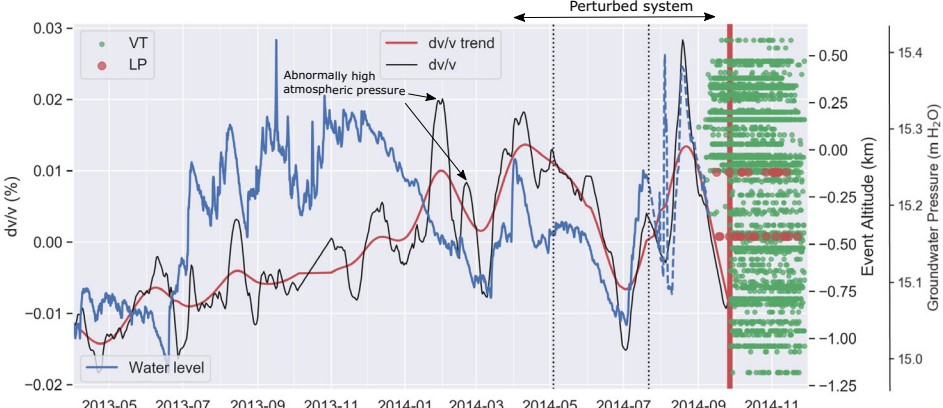

**Fig. 3 | Seismic velocity combined with complementary observations.** dv/v is estimated using daily ACF at station ONTA and inverting for a continuous velocity change time as in Brenguier et al.[30] with a weighting coefficient of 100,000 (black line). The dv/v curve has been corrected for external perturbations due to rain and snow accumulation, following Wang et al.[22]. We have also smoothed the dv/v curve using a Hodrick Prescott filter[63] with a factor of 10,000 (red line). The water level data (blue line) are from Koizumi et al.[41]. The sensor is located between MKO and MTB

stations (GOT, blue square, Fig. 1). The spikes in July and August 2014 correspond to a period attributed to sensor problems[41]. The red vertical solid line indicates the 27 September 2014 eruption, whereas black vertical dashed lines correspond to local earthquakes detected by our study. Long Period (LP) and Volcano Tectonic (VT) catalogs shown as red and green dots from August onwards and are from Zhang and Wen[44]. Rain and snow data were recorded at the station located nearby ONTN between 2012–2014 (Fig. 1) to remove the seasonal component.

observed throughout Japan by Wang et al.[22]. Although the pre-eruptive velocity decrease does not correlate with rain periods (Supplementary Fig. 11), velocity variations at the stations located on the summit fluctuate seasonally (Fig. 2 top panel). Such seasonal effects were mitigated by computing the pore pressure changes from daily precipitation and snow depth[22] (see Methods for technical details) between 2012–2014. We used seismic velocities computed following the approach developed by Brenguier et al.[30], whose linear inversion and regularization usually provide more stable results, particularly for short time series. We only used the ACF recorded before the eruption to prevent any contamination related to the eruption and post-eruptive seismic signals. We do not interpret the absolute values of the relative velocity changes using this procedure because they largely depend on smoothing coefficients.

The procedure to correct for seasonal effects accounts for changes associated with abnormal snow events but not for changes in atmospheric pressure. dv/v are sensitive to varying overlying load induced by snow thickness and atmospheric pressure, which cause an increase in seismic velocity, especially above 1 Hz[14]. Some abnormal dv/v fluctuations between January and March 2014 are coincident with

high atmospheric pressure (Supplementary Fig. 21). We find coefficients (dv/v changes divided by the load (m)) induced by atmospheric pressure) on the order of 0.1–1%/m similar to the values estimated by Donaldson et al.[14]. Spatial and temporal seismic velocity variations could also be associated with large earthquakes, as observed following the Tohoku-Oki[30] and Kumamoto[31] earthquakes, but large magnitude events have not been recorded before the 2014 Mt Ontake eruption. No seismic events larger than magnitude 4, within 100 km, were reported in the regional seismic catalogue.

A long-term increase in dv/v occurred until late April 2014 (Fig. 3). A greater velocity reduction is observed in May 2014, directly followed by a re-increase and another significant decrease (−0.04%) coincident with the onset of volcano-seismicity late August (Fig. 3, red and green dots) reported in numerous studies. The dv/v fluctuations between May and mid-August 2014 occurred during complete quiescence reported in the previous studies[6].

## Spatial constraints
The pre-eruptive velocity perturbations were restricted within a small volume on the eastern flank because they could only be

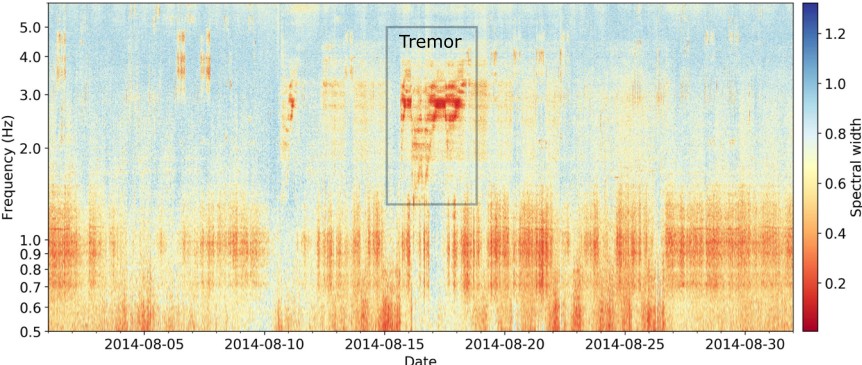

**Fig. 4 | Volcanic tremor.** Volcanic tremor detected by the network-based approach between 1.3 and 4 Hz. The width of the network covariance matrix eigenvalues distribution, called spectral width, is a proxy of the number of independent seismic sources composing the wavefield. A low value indicates a coherent signal in the network (stations used: KAD, KID, KMD, MKO, MTB, NGR, ONTA).

observed between 1–2 Hz at some sites (red triangles on Fig. 1). This is consistent with sensitivity kernels of coda waves[32,33] to seismic velocity changes that show high sensitivity close to the stations but not beyond (Fig. 1, from Supplementary Fig. 17). The absence of dv/v changes at lower frequencies (Fig. 2) corresponding to larger depths can be ascribed to the greater confining pressure that causes smaller dv/v changes[34].

The velocity variations do not depend on the lag time (Supplementary Fig. 6) but depend on the frequency band (Fig. 2), suggesting surface wave propagation rather than body wave propagation[35]. The sensitivity kernels computed for Rayleigh and Love waves suggest that the velocity changes occurred ~1–1.5 km below the station sites (Supplementary Fig. 17). These results are in line with other studies (Yamaoka et al.[6] and references therein) that estimated the source depth to less than 1–2 km beneath the crater. We also calculated the sensitivity kernels in 2D[36,37] and included diffusion and absorption effects. The kernels were computed with a velocity of 1 km/s, a mean free path of 1 km, for a time in the coda of 15 seconds, over a 500 m equally spaced grid surrounding station ONTA (Fig. 1). Changing these parameters only slightly affect the shape and total extent of the kernel.

Seismic waves are subject to strong attenuation. Contrary to ACFs, we could not detect such variations using the more traditional Cross-Correlation between station pairs approach (CC) (Supplementary Fig. 11). The coherence between CCF of station pairs is often lost depending on the interstation distance[35]. A potential problem with ACFs is the contamination of ambient noise by earthquakes and tremor[38]. To account for this problem, we also computed the phase cross-correlation (PCC[39]) using station ONTA and found the same pre-eruptive dv/v pattern (Supplementary Fig. 8).

**Complementary observations**

The most significant velocity reduction (Fig. 3) coincides with slight crustal deformation in mid-August, highlighted by retrospective stacking of deformation data[9]. Two weeks later, another slight inflation occurred[9], volcano-earthquakes around sea level were triggered[8], and stress changes were detected in the region[10]. Such activity may have opened cracks in the shallow portions of the edifice, thereby reducing seismic velocity propagation.

We applied a network-based method for detecting different kinds of seismic signals and among them volcanic tremor. The method is based on the analysis of eigenvalues of the seismic network covariance matrix[25], the Fourier-domain representation of the cross-correlation matrix. The waveforms were band-pass filtered between 0.5 and 15 Hz. The decimated (50 Hz) traces were divided into 50% overlapping time windows of which the amplitude is 1-bit normalized (independently for every station) instead of the more

common spectral whitening procedure[26]. Every pre-processed window was then subdivided into overlapping subwindows of 100 seconds, on which cross-spectral matrices are computed. From the covariance matrix eigenvalues distribution, we infer the degree of spatial coherence (spectral width, see Seydoux et al.[25]) that we can relate to the presence of seismic sources. The analysis revealed a coherent tremor between 1.3–4.0 Hz indicated by low spectral width[25] between 15 to 18 August (Fig. 4); the only of its kind in the May-August 2014 time period (Supplementary Fig. 19). We could not locate the signal with the network available. It is therefore difficult to determine the origin of these signals. Two days later, the final seismic velocity reduction started (Fig. 3). Another period of volcanic tremor was associated with the eruption (Supplementary Fig. 16) on 27 September and has been extensively described by Ogiso et al.[40].

Our retrospective work reveals unsteady behavior as early as April-May 2014 (Fig. 3). We next ran the EQTransformer pre-trained deep-learning model[27] applied in a transfer learning fashion (without retraining) on the continuous data recorded at stations KID and ONTA, and detected and picked 3265 earthquakes. We used low threshold values for the detection and picking since EQTransformer is robust to false positives. After visual inspection, most of the detections have been found consistent with earthquake signals. Two periods of enhanced seismicity have been recorded (Supplementary Fig. 15). A burst with S-P times between 3.2–3.6 seconds occurred on 22 July when the intermediate seismic velocity drop was registered (Fig. 3, black vertical dashed line). These earthquakes correspond to seismic activity ~30 km to the North of the volcano with maximum magnitudes around 4 and were directly followed by a seismic velocity decrease. A larger burst of regional earthquakes occurred on 3 May 2014 near the Yake-dake volcano, located 45 kilometers from Mt Ontake with magnitudes up to 4 when the seismic velocity started decreasing (black vertical dashed line in Fig. 3).

To further explore this intriguing seismic velocity behavior, we then investigated complementary parameters. We reviewed most studies published since the eruption (some are listed in Yamaoka et al.[6]) and could not find any clear correlation with the dv/v changes except for a pressure change recorded in GOT well (Fig. 1) located 10 km SE of the summit of the volcano[41]. The groundwater pressure is measured with a resolution of about 2 mm by a gauge in a sealed well at ~650 m depth[41]. Based on the data, Koizumi et al.[41] concluded that the 2014 eruption was not preceded by any significant magma intrusion but rather by heat transfer or small magma intrusion. The correlation between the dv/v time-series and the pressure change is striking (Fig. 3), although the pressure sensor experienced some problems between the end of July and October 2014, limiting high-frequency (~daily) comparisons between our dv/v time series and groundwater level during this period.

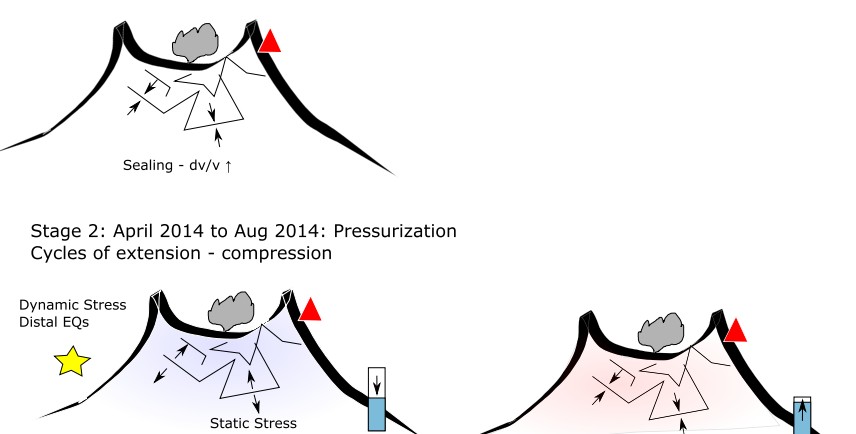

Stage 1: May 2013 to April 2014: Sealing

Sealing - dv/v ↑

Stage 2: April 2014 to Aug 2014: Pressurization
Cycles of extension - compression

Dynamic Stress
Distal EQs

Static Stress
Vol strain extension - dv/v ↓

Vol strain compression - dv/v ↑

Stage 3: August 2014 to Sept 2014: Over-pressurization

Inflation

Static Stress
Vol strain extension - dv/v ↓↓

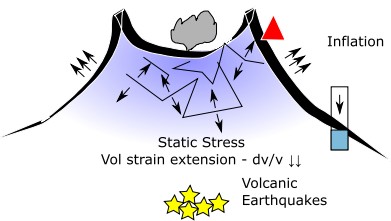

Volcanic
Earthquakes

**Fig. 5 | Conceptual model.** Conceptual model for the precursory activity prior to the 2014 eruption at Mt Ontake (not to scale). We define 3 stages corresponding to Stage 1: sealing of the sub-surface/Stage 2: pressurization/Stage 3: Over-pressurization. The red triangle corresponds to seismic stations located on the summit, yellow stars to earthquakes, blue and red colors to extensive and compressive strains, respectively, and the blue rectangle on the eastern flank to the water level in the well at GOT.

## Mechanisms driving seismic velocity changes before phreatic eruptions

Seismic velocity increased for one year, from May 2013 to May 2014. Our time series is too short to assess whether this trend was abnormal. Similar long-term dv/v trends have been observed at Whakaari/White Island volcano in the months preceding phreatic eruptions[42]. This 1-year dv/v increase may therefore indicate reduced permeability, referred to as sealing, either due to the closure of cracks or precipitation of minerals in pores[42,43] (stage 1, Fig. 5).

Shortly before the end of the increase (April 2014, stage 2), our observations include several cycles of volumetric strain variations correlated to localised dv/v changes (stage 2, Fig. 5) that indicate reversible processes. Stress sensitivity of the edifice in a linear elastic regime, density perturbation due to magma intrusion and damage accumulation beyond the linear elastic regime are three mechanisms that could explain our observations. Magma intrusion is unlikely based on other studies at Mt Ontake[8,44], whereas damage would have probably caused detectable microseismicity and deformation[45–47]. We hypothesize that the SE flank of the volcano became stressed in April 2014, following a 1-year dv/v increase.

During this period (stage 2, Fig. 5), the dv/v started responding to distal earthquakes, but only on the SE flank. The seismic velocity drop could reflect a pulse of fluid[48] or indicate that the hydrothermal system was critically pressurized through dynamic stress changes[49]. The flow of magma or a pulse of fluid from the source of earthquakes would be too slow to account for the observations[48]. Instead, this sensitivity to distal earthquakes in May and July most likely reflects the presence of highly pressurized fluids beneath the surface of Mt Ontake. Hydrological responses to earthquakes are mainly triggered by changes in permeability[50] that in turn, can affect seismic velocity[49]. As proposed by Taira et al. for geothermal reservoirs[49], ground shaking can increase apparent crack density through unclogging of fractures due to pore pressure fluctuations, thereby reducing seismic velocity. Areas of higher stress sensitivity have been found correlated with hydrothermal fluids[30] that reduce the effective confining pressure, leading to greater crack density and an increase in stress sensitivity[49].

The stress sensitivities of seismic velocity changes are in the same range as measured at other volcanoes[14,34,51]. Considering the volumetric strain sensitivity of the groundwater pressure[41], the 20 cm drop in groundwater pressure was equivalent to an increase in the volumetric strain at GOT of about 95 nstrain (0.02 m/0.0021 mm/nstrain) or assuming a bulk modulus of 18 GPa[34], a stress sensitivity $K$ of $1.7 \, e^{-7} \, Pa^{-1}$ ($K = dv/v*\varepsilon^{-1}$ here $dv/v$ is -0.02% and $\varepsilon$ is the relative strain difference[51]). These sensitivities are consistent with the range of values measured by Yamamura et al.[52]. The temporal agreement between our dv/v time series between 1–2 Hz and volumetric strain (Fig. 3) likely reflects a cycle of pressurization/depressurization in the SE sector of the volcano between April–August 2014 (stage 2, Fig. 5).

Recent numerical modelling[53] shows that volumetric strain can increase in the medium due to inflation of a spherical source 1500 meters below the surface. The volcanic edifice overall experiences extension with increased strain values close to the surface compared to the flanks. This is compatible with the absence of clear dv/v changes away from the summit between April and August 2014. These numerical simulations are also consistent with our observations whereby dv/v decreases when groundwater level decreases at GOT due to enhancement of fracture permeability, but future modelling would help clarifying this.

Several studies demonstrated the presence of over-pressurized fluids beneath the eastern flank of Mt Ontake a few days prior to the phreatic eruption[10,54]. Sano et al.[2] attributed high b-values and an increase in long-period earthquakes to the infiltration of hot fluids -10 days prior to the eruption. Our observations extend the detections of deviations of the local stress field in September highlighted by Terakawa et al.[10] using focal mechanisms. They indicate that stress changes may have actually started at least as early as April 2014, but only triggered detectable ground uplift[9] and microseismicity in late August 2014 when the fluids became over-pressurized[10] (stage 3, Fig. 5). Recent fluid injection experiments indicated that seismic velocity changes are sensitive to changes in effective stress (super-position of mechanical stress and pore pressure) producing volumetric strain while shear dislocation, therefore irreversible deformation, cannot be detected using seismic velocity[55]. This places further constraints on the processes responsible for the correlation between volumetric strain and dv/v. We postulate that our observations reflect a period of reversible but disturbed stress field induced by pressurized fluids on the eastern flank of Mt Ontake as early as April 2014. Seismic velocity could therefore be used as a proxy to detect and map the pressurized volumes caused by poroelastic effects.

The seismic velocity kept decreasing a few days after the eruption (Fig. 2) at most stations on the easter flank (Supplementary Fig. 9), consistent with a few weeks of vigorous gas emissions[56]. The cross-correlation coefficient between the reference ACF and the daily functions recovered the pre-eruptive values in December 2014 (Supplementary Fig. 4). This observation suggests that the shallow portions of the edifice were not extensively disrupted by the event, at the scale of the seismic waves, which is consistent with pressurized fluids acting as the driving force of the eruption. The absence of permanent damage in the hydrothermal system suggests that the shallow edifice could experience the same cycle of pressurization in the future.

## Perspectives and limitations

The precursory activity and driving processes for the 2014 eruption at Mt Ontake had remained hitherto unclear. This study sheds light on both aspects using single-station seismic interferometry, as previously shown for magmatic eruptions[14,29,57]. The 27 September 2014 phreatic eruption at Mt Ontake was most likely instigated by the progressive ingress of gas/steam that accumulated below the eastern crater at shallow depths (1–2 km below the crater), leading to critically stressed conditions in August 2014. Such accumulation was initiated as early as 5 months before the eruption. Seismic interferometry and its temporal resolution holds great promise for the future of volcanology as it can detect slow, early onsets of both magmatic[29] and non-magmatic events. Yet our results also show its limited spatial sensitivity above 1 Hz. Care is also needed when averaging over a network of seismic stations, especially at high frequencies[14].

The absence of clear dv/v changes outside the summit region while a volumetric strain is still detected in boreholes suggests that seismic instruments need to be installed close to the summit and/or in regions characterized by high fluid pore pressure, such as volcano-hydrothermal systems where velocity-stress sensitivity is increased. Importantly, the use of single-station seismic interferometry allows probing the shallow portions of the volcano where stress sensitivity is increased due to lower confining pressure[34].

The seismic observations presented in this study can be collected in real-time using free and open-access programs. Yet, we stress the need to use complementary observables to properly interpret dv/v results and mitigate seasonal variations, which can be challenging from a real-time monitoring perspective. Our observations extracted could then be incorporated into machine learning forecasting[5] for real-time monitoring purposes. Seismic velocity variations are increasingly being computed at volcanoes, and the model developed for Whakaari using tremor could be expanded by including dv/v[42] and the

parameters measured in this study, then tested at Mt Ontake in a transfer-learning fashion. Such models have the potential to reveal similar patterns prior to phreatic eruptions at other volcanoes worldwide. Seismic velocity increase has also been found in the months preceding phreatic eruptions at Whakaari/White Island volcano, suggesting pressurization in the edifice[42]. A comprehensive study using data recorded prior to phreatic eruptions is required to isolate distinct behaviors prior to such eruptions and discriminate between different processes; top-down processes such as sealing[58] and/or bottom-up processes such as magmatic gas influx[4]. Novel techniques such as the network-based approach to detect coherent signals or unsupervised machine-learning models, therefore, hold great potential for volcano monitoring and forecasting.

## Methods

### Data processing

A table (Supplementary Table 1) is summarizing the parameters used to present the results. Seismic records were preprocessed by carefully checking for their timing (sample alignment) and gaps (interpolating or tapering between gaps), then bandpass prefiltered between 0.01 and 8.0 Hz and finally resampled to 20 Hz. The rest of the processing followed De Plaen et al.'s[29] workflow. The seismic stations have been operated by Nagoya University (KID, KMD, MKO, MTB, MUR, TKN), JMA (ONTA, and ONTN), Gifu prefecture (GNDT), and NIED (KAD).

### Auto and cross-correlation functions

Autocorrelations (AC) were generated by cross-correlating vertical components with themselves, except for ONTN whose vertical data were not available (NN results are shown in this study). Contrary to the cross-correlation approach, the spectral whitening that sets the amplitude of the signal to 1 for all frequencies was not applied since only the phase of the signal would remain. The signals were then filtered between 1–2 Hz, which was found to be more sensitive to rainfall and volcanic events[29]. Figure 2 shows that the processing in different frequency bands (0.1–1 Hz, 0.5–1.0 Hz, and 1.0–2.0 Hz) did not provide the same dv/v. Our results agree with De Plaen et al.[29] who recovered more reliable changes between 1–2 Hz. At Mt Ontake, we generally do not detect persistent volcanic tremor between 1–2 Hz, as for example observed at Whakaari/White Island volcano[15]. Lower frequency bands did not recover the velocity reduction because they are sensitive to a larger volume below the stations and/or they are contaminated by other sources of noise (e.g., oceanic, meteorological). We ascribed such observation to spatially localized areas of velocity variations. The waveforms were then clipped to 3 rms (root-mean-square) which was found to provide the most stable results[29]. The dv/v results were only minimally influenced by this parameter (Supplementary Fig. 1). We then produced 5-day stacks (linear stacks). We opted for the 5-day stacks to present the results as it represents a good trade-off between acceptable errors and sufficient time resolution (Supplementary Fig. 2). The 1-day dv/v were for example, excessively noisy (no errors shown around the line a few days after the eruption).

### Velocity variations

The daily velocity changes were first obtained by comparing the daily AC functions to a reference AC function. Temporal variations in seismic velocity (dv/v) were derived from dt/t in the frequency domain using the Moving Window Cross Spectral Analysis (MWCS) and assuming a homogeneous velocity change. The reference used to compute the dv/v corresponded to the entire period. The post-eruption period (Supplementary Fig. 3) was not coherent enough to compute reliable dv/v estimates, as observed in Supplementary Fig. 5. We tested the approach developed by Brenguier et al.[30], who applied the MWCS analysis to daily functions and inverted for a continuous velocity change time series for each station pair separately (last sub-plot in Supplementary Fig. 3, weighting coefficient of 100,000[30]). The

results were similar (last subplot in Supplementary Fig. 3), but the computation time was much longer. The cross-correlation coefficients between the reference of the daily AC functions (between +−5 to +−35 s) were generally above 0.7 (Supplementary Fig. 4) except between the eruption and December 2014. Pre-eruptive values were only recovered in December and may indicate that the medium was not extensively disrupted by the eruption. We also processed the 2015 data, but the dv/v estimates had excessive errors (Table S1 for the parameters).

Velocity variations were then estimated between ±5 to ±35 s of the AC functions. We tested the influence of the coherence, as defined by Lecocq et al.[28], and different time lags on the final dv/v estimate. As seen in Supplementary Fig. 5 and S6, these parameters only merely affect the pre-eruptive dv/v drop. It is worth noting that low coherence values (<0.9) are observed between October and December 2014.

Other processing schemes were tested to assess the robustness of these results. The waveform stretching method[11] was used to estimate the relative velocity variations. Although seasonal variations may be slightly more pronounced with the stretching than the MWCS approach (Supplementary Fig. 7), perhaps due to changes in the source spectrum as suggested by Zhan et al.[59], dv/v estimates are very similar. Phase Cross-Correlation[39] was tested against the classical Cross-Correlation approach. This approach consists of measuring the similarity of instantaneous phases of analytical traces and as such removes the need for temporal or spectral normalization before cross-correlation. This processing provided similar results as well (Supplementary Fig. 8).

We used the vertical component of the data to compute the AC and estimate the dv/v, for example Richter et al.[35]. Autocorrelations (AC) using horizontal components and cross-component correlations (SC) were also computed but results were too scattered, and stretching results were for example unreliable. The origin of this discrepancy may be related to the enhanced sensitivity of horizontal components to different sources of noise (meteorological, anthropic) and hence, spurious dv/v estimates. Although beyond the scope of this study, further research should explore such discrepancies.

Figures S9 and S10 show the results for the individual stations located on the eastern flank and elsewhere, respectively. Results from stations not shown on these figures did not pass the quality criteria described in Table S1.

### Modeling

Rainfall data (Supplementary Fig. 12) were used to compute the pore pressure change $P(r,t)$[22]:

$$\sum_{i=1}^{n} \delta p_i \, \mathbf{erfc}\left[ r/(4c(n-i)\delta t)^{1/2} \right],$$

where $\delta t$ is the time increment from the first day $i$, $\delta_{pi}$ is the precipitation load changes ($\rho\, g\, \delta h_i$, where $\rho$ is the density (1000 kg/m³) and $g$ is the acceleration of gravity (9.81 m/s²)) at the instant $t_i$ and $c$ is the diffusion rate (m²/s). $\mathbf{erfc}$ is the complementary error function. $c = 1$ m²/s was considered as in Wang et al.[22]. The pore pressure mean value within $r = 8$ km was estimated every day, and the modeled seismic velocity change was estimated using a transfer function[22] and is shown in Supplementary Fig. 13.

Snow cover (Supplementary Fig. 12) can induce an elastic loading effect and decrease pore pressure by impeding infiltration and recharging of the groundwater[22]. These effects were considered through a linear relationship combining the effect of pore pressure and snow depth:

$$(dv/v)_{syn} = aPr + bS + C,$$

where the synthetic $(dv/v)_{syn}$ depends on $a$ and $b$, coefficients of the daily mean precipitation $Pr$ and the mean snow depth $S$, respectively,

and $C$ is a constant with offset parameters. $a = -1.488 \ast 10^{-4}$ Pa⁻¹ and $b = 2.5 \times 10^{-4}$ cm⁻¹. The results are shown in Supplementary Fig. 14.

The phase-velocity sensitivity kernels were computed for Rayleigh and Love wavesusing 1D velocity models and the surf96 program[60]. We tested the 3 models shown in Terakawa et al.[10] and consider the model of Kato et al.[8] as the most likely. All the results point to dv/v changes occurring between 0 and 2 km below the seismic station (Supplementary Fig. 17).

## Data availability

The auto correlation functions generated in this study have been deposited in the Zenodo database [add hyperlink here]. The raw seismic data are protected and are not available due to data privacy laws.

## Code availability

All the computer codes are freely and/or will be soon available (e.g., sensitivity kernels which will be a new plugin of the next version of MSNoise).

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

## Acknowledgements
The seismic data have been processed using Obspy[61]. This paper benefited from Matplotlib[62] and QGIS.org (2018) Geographic Information System. Open Source Geospatial Foundation Project. http://qgis.org. The earthquake catalogs were provided by the Japan Meteorological Agency. The water level data are from the National Institute of Advanced Industrial Science and Technology. We would like to thank two anonymous reviewers. Some constructive remarks have been very useful and improved the quality of this contribution. Nico Fournier's comments massively helped structuring the study and improved its clarity. We also wish to thank Alicia Hotovec-Ellis and Tom Winder for fruitful and insightful discussions regarding numerical modelling. The research by J.S. is supported by the projects TFassistance, financed by the Program Tenerife INNOVA 2016-2021 of the Cabildo Insular de Tenerife, and VOLRISKMAC (MAC/3.5b/124) and VOLRISKMAC II (MAC2/3.5b/328), co-financed by the INTERREG V-A Spain-Portugal MAC 2014–2020 Cooperation Program of the European Commission. C.C. acknowledged funding from the Chargé de Recherches FNRS fellowship, from the FWO (Research Foundation Flanders, Research Project G037222N entitled "Can ERT reveal the dynamics of volcanic hydrothermal systems?"), Thomas Jefferson Fund (Project "Quantifying underwater volcano degassing using novel seismo-acoustic approaches") and Campus France (Projects PHC Nusantara 47058TF and PHC Tournesol 46158UG), which greatly contributed to this study. Aspects of this study are also included in the standardization activities of the ITU/WMO/UNEP Focus Group on AI for Natural Disaster Management.

## Author contributions
Y.A. and T.T. provided and converted the data. C.C. processed the seismic data using MSNoise, Covseisnet, and EQTransformer. R.D.P. applied the PCC approach. J.S. and L.S. helped processing the data with the network-based approach. A.M. and T.L. computed the sensitivity kernels. T.L. and C.C. designed the modelling. All the authors wrote the manuscript and contributed to the interpretation.

## Competing interests
The authors declare no competing interests.
