## [Peer Review File · Nature Communications]

Hidden pressurized fluids prior to the 2014 phreatic eruption at Mt OntakeREVIEWER COMMENTS

Reviewer #1 (Remarks to the Author):

This research is an interesting addition to a well-studied eruption, the September 29, 2014 eruption of Ontake volcano, Japan. As indicated by the authors, phreatic eruptions have been difficult (if not impossible) to forecast because of a lack of timely precursory activity. Most 'precursory' signals are determined post facto through additional analyses or hindcasting. The technique presented here has clear scientific value in determining changes in velocity at active magmatic-volcanic-hydrothermal systems. But does this technique have the potential to improve forecasting? At what time period of change would the authors have suggested an impending explosion? Can the technique be applied in real-time? The authors should address these questions and make a clearer argument regarding the ability of this technique to improve explosion forecasts. Furthermore, is the signal observed at Ontake specific to that system? Or should we expect this to be seen globally or at least at systems with similar chemistry? Finally, the authors need to be specific with regards to the 'processes' that they are calling on for the velocity changes and how they are new. The technique may be new, but the idea (and evidence) of pressurized fluids within an active hydrothermal system as the driver for phreatic explosions is definitely not new. A brief discussion of the process(es) that lead to an over-pressurized system may be useful, as well. I provide specific comments and questions by section and line # below. Forecasting any eruption is the ultimate goal in volcanology, and phreatic explosions are the most challenging of phenomena. I appreciate the authors tackling this difficult issue. I hope my comments and questions improve the presentation of their results.

Initial Questions and Comments

- 1) Pressurized vs over-pressurized? Are fractures opening because of fluid pressures or changes in stress within the system caused by magma migration.
- 2) How is there change in pressurization without other detectable signals (e.g., deformation and low frequency seismicity)? What is the maximum pressure the system can achieve without deforming and generating seismicity?
- 3) Can the technique be used in real or near-realtime?
- 4) What is novel here? The technique? What are the minimum network requirements? Should the network be redesigned to improve ability to detect velocity changes?
- 5) Was there no observed increase in gas flux through crater fumaroles prior to deformation & seismicity? Or was this not measured? When observed, were the gases magmatic?
- 6) How applicable is this to other magmatic-volcanic-hydrothermal systems? At other systems with phreatic explosions, there are seismic precursors (e.g., decrease in seismicity over long- and short-time scales, for example, Telica volcano, Nicaragua).
- 7) Is steam accumulation a process? Or the end result of other processes (e.g., magma degassing, hydrothermal fluid generation and circulation)?
- 8) The authors should describe the eruption. Was the eruption one explosion? Or a series of explosions.

Title: Are the fluids 'hidden'. Were they not hypothesized to be there prior to eruption? Or are they hidden as there were no observations of degassing associated with the fluids?

Abstract:

- 1) Check number of fatalities. Varies between publications.
- 2) How much time before the 'over-pressurized' fluids deformed the system? Are they over-pressurized if they are not deforming (geodetic and seismic strain) the magmatic system?
- 3) What are the processes that were not previously known?

Introduction:

- 1) It would be good to have a short description of the eruption. Was this one explosion? Or a series of explosions?

Comments by Line

Line 19: process or processes?

Line 25: What are the previously undetected processes? Be specific. Gas accumulation has been detected or hypothesized previously. Your result extends this finding back several months. Is that a different process?

Line 32-33: WRT unambiguous signals, I would agree with the authors to some extent, however, work by Rodgers et al. (2015), Geirsson et al. (2014), and most recently Roman et al. (2016; 2019) indicate seismic precursors for phreatic explosions at Telica volcano, Nicaragua.

Line 49: Is it the origin or the forecasting that is difficult?

Line 113: How does one determine when the change initiates? In fig 3, it looks like the decreases starts in April.

Line 117: Rewrite for clarity. The quiescence was 'reported' in previous studies.

Lines 120-121: How close were the M6 earthquakes?

Line 142: triangle – triangles

Line 171: Note the frequency range of the tremor for the reader.

Line 173: Why would tremor indicate pressurization? Is not the cause of tremor often described as fluid flow?

Line 194-201: Magma intrusion can stress local and regional faults triggering distal VTs. This sentence reads as though magma is intruded along the faults, which I do not think is the case. More broadly speaking, could M4 earthquakes with sources at 45 km from the observed velocity change, change the stress field in the shallow hydrothermal system causing the velocity change (i.e., dynamic stress changes)? Is there structural control on the hydrothermal system at Ontake? Would this explain the reason for changes only observed on the eastern flank?

Lines 232-235: Were there observed changes in gas flux during the uplift and seismic episodes, or in the five months leading up to the eruption?

Would the authors have raised the alert level in June, closing the volcano off to tourists, and evacuating nearby communities?

Lines 236-240: Maybe complete the thought. The fact that one sees pre-eruptive values <2 months after the explosion, suggest that the over-pressurized fluids drove the event, and now they need to recover. Does this also suggest no permanent damage (i.e., fracturing of the hydrothermal system) that would allow for continued outgassing and an inability to pressurize?

Lines 242-243: I think previous studies cited within discuss the same 'genesis' for the phreatic eruption.

Line 249: Premises is not used correctly here.

Line 250: Is the 1 Hz threshold observed for studies at other volcanic or hydrothermal systems?

Line 295: they to 'the'

Line 302: remove the ''

Line 327: Add ',' between 'effect' and 'decrease'

Line 335: remove 'using'

Figures:

F1: The authors should show the location of the eruptive vent for the September 27, 2014 eruption.

F2: Looking at the time series of dv/v , how would one use this to forecast an eruption? It also appears as though there is change in dv/v associated with the eruption (i.e., plots of 0.5-1.0 Hz and 0.1-1.0 Hz).

F3: What does the long period of increasing dv/v (i.e., ~5/2013 – 5/2014) indicate? Lines 133-134: the dots are blue and red, not blue and green. Furthermore, the changes in dv/v observed in early 2014, appear just as large in amplitude as the changes denoted by #s 1 and 2 on the figure. Finally, the schematic model shown at the bottom of the figure is poorly described in the text. For example, what do the blue columns on the left-hand side of the figure supposed to represent?

References

Geirsson, H., Rodgers, M., LaFemina, P., dWitter, M., Roman, D., Muñoz, A., Tenorio, V., Alvarez, J., Conde Jacobo, V., Nilsson, D., Galle, B., Feineman, M.D., Furman, T., Morales, A., (2014), Multidisciplinary observations of the 2011 explosive eruption of Telica volcano, Nicaragua: Implications for the dynamics of low-explosivity ash eruptions. In: Journal of Volcanology and Geothermal Research, 271 p 55–69, doi:10.1016/j.jvolgeores.2013.11.009.

Rodgers, M., Roman, D.C., Geirsson, H., LaFemina, P., McNutt, S.R., Muñoz, A., Tenorio, V., (2015), Stable and unstable phases of elevated seismic activity at the persistently restless Telica Volcano, Nicaragua, *Journal of Volcanology and Geothermal Research*, 290, p. 63-74. doi:10.1016/j.jvolgeores.2014.11.012.

Roman, D., LaFemina, P., Bussard, R., Stephens, K., Wauthier, C., Higgins, M., Feineman, M., Arellano, S., De Moor, M., Avard, G., Martinez Cruz, M., Burton, M., Varnam, M., Saballos, A., Ibarra, M., Strauch, W., Tenorio, V., 2019, Mechanisms Of Unrest And Eruption At Persistently Restless Volcanoes: Insights From The 2015 Eruption Of Telica Volcano, Nicaragua, *Geochemistry, Geophysics, Geosystems*, v20, 8. doi: 10.1029/2019GC008450.

Roman, D., Rodgers, M., Geirsson, H., LaFemina, P., Tenorio, V., (2016), Assessing the likelihood and magnitude of volcanic explosions based on seismic quiescence. *Earth and Planetary Science Letters*, 450, 20–28. doi: 10.1016/j.epsl.2016.06.020.

Reviewer #2 (Remarks to the Author):

What are the major claims of the paper?

The authors claim to identify changes in seismic velocity occurring up to 5 months before the explosive eruption at Mt Ontake. They infer that these changes are due to pressurized fluids accumulating below the volcanic edifice. The authors suggest that they see indicators of precursory activity indicative of eruption in a time period previously thought to be quiescent.

Are they novel and will they be of interest to others in the community and the wider field? Is the work convincing, and if not, what further evidence would be required to strengthen the conclusions?

The techniques used in this paper are common to the broader ambient noise community. Pre-eruptive changes in seismic velocity that they claim to identify are commonplace observations seen preceding volcanic eruptions at other volcanoes throughout the world (e.g., Brenguier et al., 2008; Budi-Santoso & Lesage, 2016; Duputel et al., 2009; Mordret et al., 2010; Obermann et al., 2013; Bennington et al., 2018). I am also suspicious of their seismic velocity changes. Using Figure 3 as my example, dv/v is extremely noisy for the entire time period shown. I wouldn't identify any one time period as showing an increase or decrease in seismic velocity since the background trend in dv/v is so incredibly oscillatory over time. The authors show in Figure S4 that the cross-correlation coefficient associated with a fair number of their pre-eruptive changes in seismic velocity fall below 0.7. To my knowledge, all published studies use 0.7 or higher as a cut off criteria. Thus, those dv/v changes falling below 0.7 should be removed from their Figure 3 and not discussed. Many of these poorly constrained points (dv/v points w coefficient below 0.7) form the crux of their paper's argument, which is unfortunate. I am also suspicious of their removal of seasonal dv/v from their curves. They use precipitation measured from a weather station near KAD as their precipitation data for station ONTA. ONTA is at a much higher elevation atop the volcano. I would expect much different levels of precipitation there relative to near KAD. The authors note that the precipitation observed at KAD does not correlate in time w the seasonal changes in seismic velocity they observe at ONTA, which supports my claim. Thus, I am puzzled as to why they use that data to remove the seasonal velocity trend from ONTA. Since they "remove" this seasonal signal in an unfavorable way, I do not trust the results, nor the interpretation of the results, in Figure 3. Also, the authors call on different strain regimes to explain their dv/v results over time. However, they do not actually do any sort of simple strain modeling for their volcanic system. Calling upon studies that do such strain modeling at volcanoes Kilauea, for example, is not enough to justify the reason for the changes they see at an altogether different volcanic system. Also, looking at Figure 3, there are early dv/v changes that are of the same amplitude as the changes they identify in (1)-(5) of Figure 3. However, these earlier changes in dv/v are not flagged as important, nor described in the discussion. That seems like an overlooked result. Finally, the authors make the jump of linking pressurized fluids to their changes in seismic velocity. However, the evidence they give justifying this claim is extremely limited and I could

envision many other mechanisms as responsible for these changes. I would need a much expanded discussion pinpointing why these changes in seismic velocity must represent pressurization of fluids beneath the edifice.

On a more subjective note, do you feel that the paper will influence thinking in the field?

No, I feel the results from this paper are fairly similar to other ANI papers published regularly in peer reviewed journals. I also feel the paper needs a lot of work in order to present these changes in seismic velocity as robust results. As they stand, I am not sure I trust the results of Figure 3. Similarly, a lot of work needs to be done to justify the conclusions made in this paper.

Other issues

- Light grammar issues
- No discussion of the other stations or station pairs where dv/v was determined. The changes at these stations are shown in the supplement but not described in main text. This feels like an oversight.

Reviewer #3 (Remarks to the Author):

In this manuscript, the authors provide new evidences that the 2014 Ontake eruption in Japan might have been preceded by some slow pressurisation of the shallow hydrothermal system up to 5 months prior to the event. The main arguments revolve around two main lines of evidence.

First, they observed some localised changes in seismic velocity (dv/V) at a subset of monitoring stations in the 1-2 Hz band, coincident with variations of hydraulic head in a groundwater bore.

Second, they report some changes in the occurrence and nature of seismicity around the volcano in the months leading to the experience.

The results from this work are very interesting and could set some promising avenues for "symptoms to look for" in operational volcano monitoring. This is particularly important as small, seemingly unheralded non-magmatic eruptions are lethal and insofar challenging to forecast. As such, I would certainly encourage publication of these results, and Nature Communications is a suitable platform in my opinion.

A number of editorial aspects in the manuscript would need to be addressed before publication. I am not questioning the science and, instead, feel that the current manuscript does not do justice to the excellent work and results from the authors.

My main comment relates to the clarity and readability of the manuscript. I found that it currently forces to the reader to go back and forth in a number of sections, ultimately distracting from what is otherwise a nice piece of work.

The authors will find a substantial number of annotations and suggestions in the annotated manuscript attached to this review. For instance, a number of sections would benefit from changes in the order the arguments are presented. The changes in seismicity also seem to take a backstage presence in the discussion and are barely mentioned, which seems at odd with its potential importance to support the main conclusions drawn from the changes in dV/V and pressure head.

The figures are generally well done and informative. One that I would quite strongly recommend to adjust is Fig 3. My suggestion would be to focus the figure on the long term dv/V , and split the conceptual model into a separate, later figure. That separate figure could then focus on the conceptual models, and maybe include a subset of the dv/V and pressure head timeseries zooming into the shorter time period where there two dataset exists (e.g., ~ March 2014 to eruption).

As it currently stands, the figure is confusing and present models and dataset which are only

discussed much later in the manuscript. The main consequences are some confusion when the figure is first presented and, later down the manuscript, some back and forth between the discussion and that figure.

Overall, this manuscript is well-worth publishing in Nature Communications once the edits provide a clearer, easier to follow, and ultimately more comprehensive and citable paper.

I remain available to the authors and the editor should any clarifications on the review be needed.

Kind regards,

Nico

Nico Fournier
GNS Science
Taupō
Aotearoa New Zealand

REVIEWER COMMENTS

Reviewer #1 (Remarks to the Author)

This research is an interesting addition to a well-studied eruption, the September 29, 2014 eruption of Ontake volcano, Japan. As indicated by the authors, phreatic eruptions have been difficult (if not impossible) to forecast because of a lack of timely precursory activity. Most ‘precursory’ signals are determined post facto through additional analyses or hindcasting. The technique presented here has clear scientific value in determining changes in velocity at active magmatic-volcanic-hydrothermal systems. But does this technique have the potential to improve forecasting? At what time period of change would the authors have suggested an impending explosion? Can the technique be applied in real-time? The authors should address these questions and make a clearer argument regarding the ability of this technique to improve explosion forecasts. Furthermore, is the signal observed at Ontake specific to that system? Or should we expect this to be seen globally or at least at systems with similar chemistry?

→ *Thanks for raising these questions. We reply below to each specific question*

R1-1 But does this technique have the potential to improve forecasting?

→ *dv/v shows up earlier than any other seismic-related observation already reported and offer critical elements of interpretation for a potential future unrest. This insight into the volcanic processes preceding the eruption becomes clear in the light of the variations of seismic velocity we observe, specifically when compared with what other observations would offer both in isolation and in real time.*

We are convinced that these approaches could improve forecasting. Different open-source softwares have been developed by some of us and are already running in real-time at several volcano observatories and sometimes support decision-making. Our contribution shows new directions to forecast phreatic eruptions that might be useful in the future.

R1-2 At what time period of change would the authors have suggested an impending explosion

→ *We cannot predict a specific “count down” to the explosion. Rather, we identify evolving signs indicating an increasing probability for unrest/the system reaching a critical state. That is, given that we have simultaneous access to all the observables.*

→ *We would have been more vigilant when the first significant drop was noticed coincident with distal earthquake activity and even more concerned in August 2014 when one of our approaches detected some tremor (mid-August). Yet for obvious political/diplomatical reasons, we do not make such statement in the main text.*

R1-3 Can the technique be applied in real-time?

→ *Yes the dv/v can be computed using the MSNoise software, the spectral width using Covseisnet and the automatic detection of seismic events using EQTransformer. The dv/v correction using meteorological data would require an extra step. However, we think that the most promising way to forecast phreatic eruptions in real-time is to incorporate these observations into a forecasting model such as developed by Dempsey et al. (2020) and Ardid et al. (accepted)*

R1-4 Furthermore, is the signal observed at Ontake specific to that system? Or should we expect this to be seen globally or at least at systems with similar chemistry?

→ *We don't think so and hope to find similar types of signals at other systems. We have already processed several dataset showing similar results (e.g., Whakaari/White Island eruption in 2019 (Caudron et al., 2021, Earth Planets and Space). We have added a new paragraph discussing these aspects at the end of the manuscript*

R1-5 Finally, the authors need to be specific with regards to the ‘processes’ that they are calling on for the velocity changes and how they are new. The technique may be new, but the idea (and evidence) of

pressurized fluids within an active hydrothermal system as the driver for phreatic explosions is definitely not new. A brief discussion of the process(es) that lead to an over-pressurized system may be useful, as well. I provide specific comments and questions by section and line # below. Forecasting any eruption is the ultimate goal in volcanology, and phreatic explosions are the most challenging of phenomena. I appreciate the authors tackling this difficult issue. I hope my comments and questions improve the presentation of their results.

➔ *This is a fair comment and we thank you for raising these questions that have triggered new research and interpretation. Pressurized fluids have been proposed as trigger for phreatic eruptions. In the case of this very well-studied eruption, most studies had detected changes up to 1 month prior to the event. Our study shows that some section of the volcano may have actually been pressurized before that coincident with changes of strain identified on the SE flank. We further suggest that this is corroborated by dv/v response to regional earthquakes. We are now providing a dedicated section to the pre-eruptive processes. We have now written a section entitled “Mechanisms driving seismic velocity changes before phreatic eruptions” in lines 233-300 and have made a new figure.*

Initial Questions and Comments

1) Pressurized vs over-pressurized? Are fractures opening because of fluid pressures or changes in stress within the system caused by magma migration.

➔ *Thank you for bringing this point. We know that no deformation was detected before late August 2014. Therefore, we decided to distinguish between pressurized (stage 2) over-pressurized (stage 3).*

➔ *We do not have any evidence for the opening of fractures triggering detectable microseismicity before the end of August. This of course does not mean that no fractures opened/closed. In light of the volumetric strain results, we think that we are sensitive to stress changes triggering tiny but detectable dv/v changes and changes in water pressure on the eastern flank. No magma migration was detected prior to the 2014 eruption, excepting the 10-min right before it occurred. We have now dedicated an entire section (Mechanisms driving seismic velocity changes before phreatic eruptions” in lines 233-300) to interpret our findings*

2) How is there change in pressurization without other detectable signals (e.g., deformation and low frequency seismicity)? What is the maximum pressure the system can achieve without deforming and generating seismicity?

➔ *This is one of the advantage of this technique compared to earthquake or ground deformation, as show in one of the pioneering study (Brenguier et al., 2008; Nat Geo), besides its temporal resolution as a continuous monitoring approach. The relation between shallow pressure, deformation at the surface and seismicity should be explored using dedicated numerical modeling. Some collaborators and an ongoing PhD thesis are exploring this in detail. We have added a paragraph to discuss this in detail (lines 270-276):*
“Recent numerical modelling⁵⁴ show that volumetric strain increases in the medium due to inflation of a spherical source 1500 meters below the surface. The volcanic edifice experiences extension with increased strain values close to the surface compared to the flanks. This is compatible with the absence of clear dv/v changes away from the summit between April and August 2014. These numerical simulations are also consistent with our observations whereby dv/v decreases when groundwater level decreases at GOT due to enhancement of fracture permeability.”

3) Can the technique be used in real or near-realtime?

➔ *Absolutely, please see RI-3*

4) What is novel here? The technique? What are the minimum network requirements? Should the network be redesigned to improve ability to detect velocity changes?

- ➔ *The dv/v computations are not novel. We have explored various unconventional data processing approaches (PCC, no reference) and have accounted for meteorological effects following Wang et al.'s study. We have used relatively novel techniques (Covseisnet and EQTransformer) to detect and study the earthquakes, including a volcanic tremor previously undetected in August 2014. Our findings bring new light into this important eruption (see R1-5)*
- ➔ *One of the messages of our study concerns the importance of stations lying on top of hydrothermal systems. They may be more sensitive to external and internal disturbances and could therefore provide critical information*

5) Was there no observed increase in gas flux through crater fumaroles prior to deformation & seismicity? Or was this not measured? When observed, were the gases magmatic?

- ➔ *No there is no reported change in gas flux through crater fumaroles*

6) How applicable is this to other magmatic-volcanic-hydrothermal systems? At other systems with phreatic explosions, there are seismic precursors (e.g., decrease in seismicity over long- and short-time scales, for example, Telica volcano, Nicaragua).

- ➔ *This can be applied at any other systems, as long as there is a seismic station, ideally on top of the volcano. Telica is one of the datasets we have starting exploring with promising results. This will be the topic of a more comprehensive study where we will apply the same data processing procedure and parameters to various volcano-seismic datasets.*

7) Is steam accumulation a process? Or the end result of other processes (e.g., magma degassing, hydrothermal fluid generation and circulation)?

- ➔ *Again, this is a good point. We have now revisited our interpretation in light of the new results and the volumetric strain (see R1-5)*

8) The authors should describe the eruption. Was the eruption one explosion? Or a series of explosions.

- ➔ *Done. Please see lines 55-58. Although there is a large amount of papers published, do not hesitate if you would like us to provide more information*

Title: Are the fluids 'hidden'. Were they not hypothesized to be there prior to eruption? Or are they hidden as there were no observations of degassing associated with the fluids?

- ➔ *Terakawa et al. (2016) had suggested the injection of magmatic gas into the hydrothermal before the 2014 eruption based on deviations of the focal mechanisms in the east flank for a week before the 2014 eruption. Our results, based on continuous noise show that this situation may actually have started several months before this. This is the first observation indicating pressurization in the shallow edifice as early as April 2014. This is why we have used the term hidden. Note that we have replaced over-pressurized by pressurized following the Reviewer's comment.*

Abstract

1) Check number of fatalities. Varies between publications.

- ➔ *We have used the number reported in the introduction of the Mt Ontake special issue, i.e., at least 58 hikers. But the numbers vary even within the special issue*

2) How much time before the 'over-pressurized' fluids deformed the system? Are they over-pressurized if they are not deforming (geodetic and seismic strain) the magmatic system?

- ➔ *Again, thanks for this question. We think that this applies to the shallow volcanic edifice, likely not the magmatic system, since no magma appeared to be involved. The reviewer is right that the fluids were probably pressurized until late August then over-pressurized until the eruption when significant seismicity, as well as detectable deformation, were observed. We have now clarified this (see R1-5).*

3) What are the processes that were not previously known?

➔ *We uncover pressurized fluids and explore possible processes to explain our observations*

Introduction:

1) It would be good to have a short description of the eruption. Was this one explosion? Or a series of explosions?

➔ *Suggestion followed.*

Comments by Line

Line 19: process or processes?

➔ *Processes. Corrected*

Line 25: What are the previously undetected processes? Be specific. Gas accumulation has been detected or hypothesized previously. Your result extends this finding back several months. Is that a different process?

➔ *Gas accumulation had only been hypothesized to have started a few weeks before the eruption. We have changed this sentence to be more specific (Lines 27-29):*

“Our results shed light onto previously undetected pressurized fluids using stations located above the volcano-hydrothermal system and holds great potential for monitoring.”

Line 32-33: WRT unambiguous signals, I would agree with the authors to some extent, however, work by Rodgers et al. (2015), Geirsson et al. (2014), and most recently Roman et al. (2016; 2019) indicate seismic precursors for phreatic explosions at Telica volcano, Nicaragua.

➔ *Again, we agree with the reviewer and know these papers very well. But as some of our studies on phreatic eruptions, these approaches are useful to identify long-term (Rodgers et al., 2016) or very short-term changes (the seismic quiescence paper) that are very useful to detect unrest. Similarly, we have found some changes in amplitude ratios (Caudron et al., 2019, Geology) or VLPs shortly before eruptions (Caudron et al., 2018, EPS). All these signals can be considered as precursors. The most promising approach, in our opinion, has been proposed by Dempsey et al. (2020) who designed a forecaster. We have modified the sentence accordingly (Lines 36-37):*

“The absence of forecasting signals challenges volcanologists' knowledge”

Line 49: Is it the origin or the forecasting that is difficult?

➔ *The forecasting and precursory processes. We have rephrased (Lines 55-56):*

“The forecasting and precursory processes of this eruption, therefore, remain difficult to untangle”

Line 113: How does one determine when the change initiates? In fig 3, it looks like the decreases starts in April.

➔ *The reviewer was right. We have completely revisited the chronology and interpretation in light of the new results (see R1-5).*

Line 117: Rewrite for clarity. The quiescence was ‘reported’ in previous studies.

➔ *corrected*

Lines 120-121: How close were the M6 earthquakes?

➔ *The largest earthquakes in the catalogs were M4 earthquakes located 45 km away from the volcano, not M6. We have modified the text accordingly (Lines 214-219):*

“These earthquakes correspond to seismic activity ~30 km to the North of the volcano with maximum magnitudes around 4 and were directly followed by a seismic velocity decrease. A larger burst of regional earthquakes occurred on 3 May 2014 near the Yakedake volcano,

located 45 kilometers from Mt Ontake with magnitudes up to 4 when the seismic velocity started decreasing (black vertical dashed line in Fig. 3)."

Line 142: triangle – triangles

→ *corrected*

Line 171: Note the frequency range of the tremor for the reader.

→ *added*

Line 173: Why would tremor indicate pressurization? Is not the cause of tremor often described as fluid flow?

→ *It can be both. Since we could not locate the tremor reliably, we have decided no to elaborate further. Lines 196-197:*

"We could not locate the signal with the network available. It is therefore difficult to determine the origin of this signals."

Line 194-201: Magma intrusion can stress local and regional faults triggering distal VTs. This sentence reads as though magma is intruded along the faults, which I do not think is the case. More broadly speaking, could M4 earthquakes with sources at 45 km from the observed velocity change, change the stress field in the shallow hydrothermal system causing the velocity change (i.e., dynamic stress changes)? Is there structural control on the hydrothermal system at Ontake? Would this explain the reason for changes only observed on the eastern flank?

→ *We haven't found any study documenting structural control unfortunately. This would have been interesting. Together with the strain data derived from the nearby well, the recent numerical modeling results of Hotovec-Ellis (in press) are now used to refine our interpretation (Lines 270-276)*

"Recent numerical modelling⁵⁴ show that volumetric strain increases in the medium due to inflation of a spherical source 1500 meters below the surface. The volcanic edifice experiences extension with increased strain values close to the surface compared to the flanks. This is compatible with the absence of clear dv/v changes away from the summit between April and August 2014. These numerical simulations are also consistent with our observations whereby dv/v decreases when groundwater level decreases at GOT due to enhancement of fracture permeability."

→ *Taira et al. 2018 (Science Advances) have explored the relation between dv/v and dynamic stress transients. They have found sudden velocity reductions caused by openings of fractures triggered by small local and regional earthquakes (as small as 0.08 MPa). We have now dedicated two paragraphs to discuss this aspect (Lines 247-279):*

"During this period (stage 2, Fig. 4), the dv/v started responding to distal earthquakes, but only on the SE flank. The seismic velocity drop could reflect a pulse of fluid⁴⁹ or indicate that the hydrothermal system was critically pressurized through dynamic stress changes⁵⁰. The flow of magma or a pulse of fluid from the source of earthquakes would be too slow to account for the observations⁴⁹. Instead, this sensitivity to distal earthquakes in May and July most likely reflects the presence of highly pressurized fluids beneath the surface of Mt Ontake. Hydrological responses to earthquakes are mainly triggered by changes in permeability⁵¹ that in turn can affect seismic velocity⁵⁰. As proposed by Taira et al. for geothermal reservoirs⁵⁰, ground shaking can increase apparent crack density through unclogging of fractures due to pore pressure fluctuations, thereby reducing seismic velocity. Areas of higher stress sensitivity have been found correlated with hydrothermal fluids³⁰ that reduce the effective confining pressure, leading to greater crack density and increase of stress sensitivity⁵⁰.

The stress sensitivities of seismic velocity changes are in the same range as measured at other volcanoes^{37,14,52}. Considering the volumetric strain sensitivity of the groundwater pressure³³, the 20 cm drop in groundwater pressure was equivalent to an increase in the volumetric strain

at GOT of about 95 nstrain (0.02 m/0.0021 mm/nstrain) or assuming a bulk modulus of 18 GPa³⁷, a stress sensitivity K of $1.7 \text{ e}^{-7} \text{ Pa}^{-1}$ ($K=dv/v*\epsilon^{-1}$ here dv/v is $\sim 0.02\%$ and ϵ is the relative strain difference⁵²). These sensitivities are consistent with the range of values measured by Yamamura et al.⁵³. The temporal agreement between our dv/v time series between 1-2 Hz and volumetric strain (Fig. 3) likely reflects a cycle of pressurization/depressurization in the SE sector of the volcano between April-August 2014 (stage 2, Fig. 5).”

Lines 232-235: Were there observed changes in gas flux during the uplift and seismic episodes, or in the five months leading up to the eruption?

→ *No*

Would the authors have raised the alert level in June, closing the volcano off to tourists, and evacuating nearby communities?

→ *Based on our observations, we may have raised the alert level in June and closed the volcano when tremor was detected in August. But we do not want to add this in the manuscript.*

Lines 236-240: Maybe complete the thought. The fact that one sees pre-eruptive values <2 months after the explosion, suggest that the over-pressurized fluids drove the event, and now they need to recover. Does this also suggest no permanent damage (i.e., fracturing of the hydrothermal system) that would allow for continued outgassing and an inability to pressurize?

→ *Thanks again for this suggestion. We have now mentioned this (lines 293-300).*

“The seismic velocity kept decreasing a few days after the eruption (Fig. 2) at most stations on the easter flank (Fig. S9) consistent with a few weeks of vigorous gas emissions⁵⁷. The cross-correlation coefficient between the reference ACF and the daily functions recovered the pre-eruptive values in December 2014 (Fig. S4). This observation suggests that the shallow portions of the edifice were not extensively disrupted by the event, at the scale of the seismic waves, which is consistent with pressurized fluids acting as the driving force of the eruption. The absence of permanent damage in the hydrothermal system suggests that the shallow edifice could experience the same cycle of pressurization in the future.”

Lines 242-243: I think previous studies cited within discuss the same ‘genesis’ for the phreatic eruption.

→ *Yes several studies have postulated that (over-)pressurised fluids may drive phreatic eruptions. The situation and timing at Mt Ontake remained unclear. We provide new observations and interpretation.*

Line 249: Premises is not used correctly here.

→ *corrected*

Line 250: Is the 1 Hz threshold observed for studies at other volcanic or hydrothermal systems?

→ *Yes most often, we have found better SNR above 1 Hz for the single station approach, not for the station pairs approach*

Line 295: they to ‘the’

→ *corrected*

Line 302: remove the ‘)’

→ *corrected*

Line 327: Add ‘,’ between ‘effect’ and ‘decrease’

→ *corrected*

Line 335: remove 'using'

→ *corrected*

Figures

F1: The authors should show the location of the eruptive vent for the September 27, 2014 eruption.

→ done

F2: Looking at the time series of dv/v , how would one use this to forecast an eruption? It also appears as though there is change in dv/v associated with the eruption (i.e., plots of 0.5-1.0 Hz and 0.1-1.0 Hz).

→ *With the single station approach, the changes at lower frequencies have always been noisier and prone to errors. Although we process the data at low frequencies we typically only use the 1-2 Hz results for real-time monitoring using the single station approach. This being said, there is indeed a drop associated with the eruption itself. Overall, we think that these dv/v observations need to be complemented by other observations and used in robust forecasting tools.*

F3: What does the long period of increasing dv/v (i.e., ~5/2013 – 5/2014) indicate? Lines

→ *Thanks again for pointing this. We think it may represent sealing. We now explain and discuss this but insist on the fact that it is speculative with only 2 years of data (lines 233-238) "Seismic velocity increased for one year, from May 2013 to May 2014. Our timeseries is too short to assess whether this trend was abnormal. Similar long-term dv/v trends have been observed at Whakaari/White Island volcano in the months preceding phreatic eruptions⁴⁴. This 1-year dv/v increase may therefore indicate reduced permeability, referred to as sealing, either due to the closure of cracks or precipitation of minerals in pores^{44,45} (stage 1, Fig. 5)."*

133-134: the dots are blue and red, not blue and green. Furthermore, the changes in dv/v observed in early 2014, appear just as large in amplitude as the changes denoted by #s 1 and 2 on the figure.

→ *Yes, we have investigated these changes and they actually relate to periods of greater atmospheric pressure*

Finally, the schematic model shown at the bottom of the figure is poorly described in the text. For example, what do the blue columns on the left-hand side of the figure supposed to represent?

→ *This has been completely revised following the Reviewers' comments (see R1-5)*

Reviewer #2 (Remarks to the Author):

The techniques used in this paper are common to the broader ambient noise community. Pre-eruptive changes in seismic velocity that they claim to identify are commonplace observations seen preceding volcanic eruptions at other volcanoes throughout the world (e.g., Brenguier et al., 2008; Budi-Santoso & Lesage, 2016; Duputel et al., 2009; Mordret et al., 2010; Obermann et al., 2013; Bennington et al., 2018). I am also suspicious of their seismic velocity changes. Using Figure 3 as my example, dv/v is extremely noisy for the entire time period shown. I wouldn't identify any one time period as showing an increase or decrease in seismic velocity since the background trend in dv/v is so incredibly oscillatory over time.

→ *Could the reviewer be more specific? To the best of our knowledge, we have tested most approaches currently available within the community that overall provide comparable results. There are of course differences as you could expect when you estimate dv/v in the time or the frequency domain. We have also used the PCC approach which is very rarely used in the community. The reviewer has rightly pointed an oscillation in January 2014 which led us to even more carefully explore this behavior.*

The authors show in Figure S4 that the cross-correlation coefficient associated with a fair number of their pre-eruptive changes in seismic velocity fall below 0.7. To my knowledge, all published studies use 0.7 or higher as a cut off criteria. Thus, those dv/v changes falling below 0.7 should be removed from their Figure 3 and not discussed. Many of these poorly constrained points (dv/v points w coefficient below 0.7) form the crux of their paper's argument, which is unfortunate.

→ *A cross correlation coefficient is definitely not the best, nor the only, way to assess an error. As the reviewer likely knows, a sudden drop of cross correlation coefficients can be related to sudden ruptures (faults). We also would like to stress that the use of a static reference is definitely not ideal. This is why we have used a relatively new approach that does not take any reference into account extending the approach developed by Brenguier et al. (2014). But as pointed above, we would pleased to test one more approach if the reviewer has any suggestion and any code to share.*

I am also suspicious of their removal of seasonal dv/v from their curves. They use precipitation measured from a weather station near KAD as their precipitation data for station ONTA. ONTA is at a much higher elevation atop the volcano. I would expect much different levels of precipitation there relative to near KAD. The authors note that the precipitation observed at KAD does not correlate in time w the seasonal changes in seismic velocity they observe at ONTA, which supports my claim. Thus, I am puzzled as to why they use that data to remove the seasonal velocity trend from ONTA. Since they "remove" this seasonal signal in an unfavorable way, I do not trust the results, nor the interpretation of the results, in Figure 3.

→ *We apologize for the confusion. The rain/snow were not measured close to station KAD as mentioned earlier in line 105 ("recorded nearby station KAD") and plotted as a purple square in Figure 1 (but not very visible). The station was actually located next to ONTA/ONTN as mentioned in Figure S12, 'Caption Figure S12. Rain (blue line) and snow (black) measured at the station co-located with ONTA (Figure 3).'*

→ *It is quite fortunate and rare to have the opportunity to use a meteorological station co-located with a seismic one.*

Also, the authors call on different strain regimes to explain their dv/v results over time. However, they do not actually do any sort of simple strain modeling for their volcanic system. Calling upon studies that do such strain modeling at volcanoes Kilauea, for example, is not enough to justify the reason for the changes they see at an altogether different volcanic system. Also, looking at Figure 3, there are early dv/v changes that are of the same amplitude as the changes they identify in (1)-(5) of Figure 3. However, these earlier changes in dv/v are not flagged as important, nor described in the discussion. That seems like an overlooked result.

→ *Thanks for pointing these. We have now added a red curve with less dramatic changes over time. We think it's useful but can be slightly misleading for real-time forecasting purposes.*

This being said, we have explored new complementary dataset which have been provided by different co-authors.

Finally, the authors make the jump of linking pressurized fluids to their changes in seismic velocity. However, the evidence they give justifying this claim is extremely limited and I could envision many other mechanisms as responsible for these changes. I would need a much expanded discussion pinpointing why these changes in seismic velocity must represent pressurization of fluids beneath the edifice.

➔ *We have expanded the discussion massively (see R1-5) and would be very happy to know which mechanisms do the reviewer envisage for these kind of eruptions. The main triggering processes for this eruption, but also similar phreatic eruptions, have been considered. We have reviewed them in Caudron et al. (2021) (<https://earth-planet-space.springeropen.com/articles/10.1186/s40623-021-01506-0>) but are of course willing to consider any other mechanism.*

On a more subjective note, do you feel that the paper will influence thinking in the field?

No, I feel the results from this paper are fairly similar to other ANI papers published regularly in peer reviewed journals. I also feel the paper needs a lot of work in order to present these changes in seismic velocity as robust results. As they stand, I am not sure I trust the results of Figure 3. Similarly, a lot of work needs to be done to justify the conclusions made in this paper.

➔ *We thank the reviewer for her/his comments. We disagree with this statement and have tried to address her/his question, but would be happy to answer and explore any other constructive comment.*

Other issues

- Light grammar issues

➔ *Could the reviewer be more specific?*

- No discussion of the other stations or station pairs where dv/v was determined. The changes at these stations are shown in the supplement but not described in main text. This feels like an oversight.

➔ *We have added 12 figures in the supplementary material to describe all the tests that we have made. Hence we have the feeling to have provided quite transparent results.*

From the pdf

I don't see shaded errors on F2

➔ *It's there but the errors are small*

dv/v caused by earthquakes happens over a much shorter timescale than what you show, so unlikely to be caused by tectonics.

➔ *Yes the co-seismic drops are sudden indeed but the final dv/v are smoothed during the inversion.*

How does strain compare to earlier dv/v changes? why only plot that smaller time series?

➔ *We have now investigated this in detail. Please see figure 3 and our response to reviewer 1 (R1-5)*

What do the coda look like to promote or refute this claim? I need a stronger link to coda being primarily surface waves before I accept the next statement using sensitivity kernels to constrain the depth of changes

➔ *This is one of the most straightforward way to discriminate between body and surface waves. We would like to direct the reviewer to the pioneer published by Oberman et al in 2014 and extensively referenced in the literature. Again, we are more than happy to explore any other method the reviewer would indicate.*

What kind of deformation- compaction? Extension?

➔ *Inflation. Please see the reference cited in our manuscript*

Why is this the likely inference? I need more justification to this claim

→ *Volcanic tremor can reflect pressurization or the movement of fluids. We cannot be more specific with our current results. We could not get reliable tremor locations.*

Simple strain modeling of the system at this volcano should be carried out to demonstrate this.

Packages like Coulomb 3.3 are freely available for such simple modeling

→ *We have extensively discussed this with two collaborators, including Alicia Hotovec-Ellis who recently explored this in detail:*

<https://agupubs.onlinelibrary.wiley.com/doi/full/10.1029/2021JB023324>. Although we have used Coulomb in the past (Donaldson et al. 2017; 2019), there seems to be critical issues associated with this software.

Reviewer #3 (Remarks to the Author):

In this manuscript, the authors provide new evidences that the 2014 Ontake eruption in Japan might have been preceded by some slow pressurisation of the shallow hydrothermal system up to 5 months prior to the event. The main arguments revolve around two main lines of evidence.

First, they observed some localised changes in seismic velocity (dv/V) at a subset of monitoring stations in the 1-2 Hz band, coincident with variations of hydraulic head in a groundwater bore.

Second, they report some changes in the occurrence and nature of seismicity around the volcano in the months leading to the experience.

The results from this work are very interesting and could set some promising avenues for "symptoms to look for" in operational volcano monitoring. This is particularly important as small, seemingly unheralded non-magmatic eruptions are lethal and insofar challenging to forecast. As such, I would certainly encourage publication of these results, and Nature Communications is a suitable platform in my opinion.

A number of editorial aspects in the manuscript would need to be addressed before publication. I am not questioning the science and, instead, feel that the current manuscript does not do justice to the excellent work and results from the authors.

My main comment relates to the clarity and readability of the manuscript. I found that it currently forces to the reader to go back and forth in a number of sections, ultimately distracting from what is otherwise a nice piece of work.

The authors will find a substantial number of annotations and suggestions in the annotated manuscript attached to this review. For instance, a number of sections would benefit from changes in the order the arguments are presented. The changes in seismicity also seem to take a backstage presence in the discussion and are barely mentioned, which seems at odd with its potential importance to support the main conclusions drawn from the changes in dV/V and pressure head.

The figures are generally well done and informative. One that I would quite strongly recommend to adjust is Fig 3. My suggestion would be to focus the figure on the long term dv/V , and split the conceptual model into a separate, later figure. That separate figure could then focus on the conceptual models, and maybe include a subset of the dv/V and pressure head timeseries zooming into the shorter time period where there two dataset exists (e.g., ~ March 2014 to eruption).

As it currently stands, the figure is confusing and present models and dataset which are only discussed much later in the manuscript. The main consequences are some confusion when the figure is first presented and, later down the manuscript, some back and forth between the discussion and that figure.

Overall, this manuscript is well-worth publishing in Nature Communications once the edits provide a clearer, easier to follow, and ultimately more comprehensive and citable paper.

I remain available to the authors and the editor should any clarifications on the review be needed.

Kind regards,

Nico

Nico Fournier

We would sincerely like to thank Nico Fournier. His remarks have been extremely useful and have triggered an in-depth revision of the interpretation of this iconic eruption. We have followed all the suggestions he has provided in his pdf and would be more than happy to read his suggestions regarding this new version

Regarding the question at the bottom of page 11: 'Open systems?'

→ we have now examined what happened before and after in terms of degassing and provide an interpretation using the cross correlation coefficient (lines XX)

“The cross-correlation coefficient between the reference ACF and the daily functions recovered the pre-eruptive values in December 2014 (Figure S4). This observation suggests that the shallow portions of the edifice were not extensively disrupted by the event, at the scale of the seismic waves, which is consistent with pressurized fluids acting as the driving force of the eruption. The absence of permanent damage in the hydrothermal system suggests that the shallow edifice could experience the same cycle of pressurization in the future.”

→ We do not compare our results with other systems as data processing is different in each study. A PhD student is currently exploring different volcano seismic dataset using the same data processing parameters.

REVIEWERS' COMMENTS

Reviewer #2 (Remarks to the Author):

The authors have addressed all of my comments in detail and I am comfortable with accepting the revised manuscript in its current form for publication.